# Anti-Inflammatory CDGSH Iron-Sulfur Domain 2: A Biomarker of Central Nervous System Insult in Cellular, Animal Models and Patients

**DOI:** 10.3390/biomedicines10040777

**Published:** 2022-03-27

**Authors:** Woon-Man Kung, Chai-Ching Lin, Wei-Jung Chen, Li-Lin Jiang, Yu-Yo Sun, Kuang-Hui Hsieh, Muh-Shi Lin

**Affiliations:** 1Division of Neurosurgery, Department of Surgery, Taipei Tzu Chi Hospital, Buddhist Tzu Chi Medical Foundation, New Taipei City 23142, Taiwan; nskungwm@yahoo.com.tw; 2Department of Exercise and Health Promotion, College of Kinesiology and Health, Chinese Culture University, Taipei 11114, Taiwan; 3Department of Biotechnology and Animal Science, College of Bioresources, National Ilan University, Yilan 26047, Taiwan; lincc@niu.edu.tw (C.-C.L.); wjchen@niu.edu.tw (W.-J.C.); piger2032@hotmail.com (L.-L.J.); 4Institute of Biopharmaceutical Sciences, National Sun Yat-sen University, Kaohsiung 804201, Taiwan; yuyosun42@gmail.com; 5Department of Neuroscience, Center for Brain Immunology and Glia (BIG), University of Virginia School of Medicine, Charlottesville, VA 22903, USA; 6Department of Laboratory Service, Kuang Tien General Hospital, Taichung 43303, Taiwan; hsiehkuanghui@gmail.com; 7Division of Neurosurgery, Department of Surgery, Kuang Tien General Hospital, Taichung 43303, Taiwan; 8Department of Biotechnology, College of Medical and Health Care, Hung Kuang University, Taichung 43302, Taiwan; 9Department of Health Business Administration, College of Medical and Health Care, Hung Kuang University, Taichung 43302, Taiwan

**Keywords:** CISD2, CNS insult biomarker, anti-inflammatory effect, therapeutic target

## Abstract

Spinal cord injury (SCI) promotes brain inflammation; conversely, brain injury promotes spinal neuron loss. There is a need to identify molecular biomarkers and therapeutic targets for central nervous system (CNS) injury. CDGSH iron-sulfur structural domain 2 (CISD2), an NF-κB antagonist, is downregulated after injury in vivo and in vitro. We aimed to examine the diagnostic value of CISD2 in patients with CNS insult. Plasma and cerebrospinal fluid (CSF) CISD2 levels were decreased in 13 patients with CNS insult and were negatively correlated with plasma IL6 levels (associated with disease severity; r = −0.7062; *p* < 0.01). SCI-induced inflammatory mediators delivered through CSF promoted mouse brain inflammation at 1 h post-SCI. Anti-CISD2 antibody treatment exacerbated SCI-induced inflammation in mouse spine and brain. Lipopolysaccharide-stimulated siCISD2-transfected EOC microglial cells exhibited proinflammatory phenotypes (enhanced M1 polarization, decreased M2 polarization, and increased intranuclear NF-κB p65 translocation). Plasma and CSF CISD2 levels were increased in three patients with CNS insult post-therapeutic hypothermia. CISD2 levels were negatively correlated with plasma and CSF levels of inflammatory mediators. CISD2 inhibition and potentiation experiments in cells, animals, and humans revealed CISD2 as a biomarker for CNS insult and upregulation of CISD2 anti-inflammatory properties as a potential therapeutic strategy for CNS insult.

## 1. Introduction

High-grade and low-grade central nervous system (CNS) insults, such as CNS injuries (including traumatic brain injuries (TBIs), spinal cord injuries (SCIs), and hemorrhagic/ischemic stroke) and CNS diseases (including neurodegenerative diseases and neoplasms) [1] promote the pathological activation of microglia and neuroinflammation. Microglial cells are the resident immune cells of the CNS [2]. The inflammatory cascades activated by microglial cells promote mitochondrial dysfunction [3,4] and the production of reactive oxygen species (ROS), which further augment proinflammatory responses [5]. Pathological neuroinflammation and mitochondrial dysfunction regulate each other and promote irreversible neuronal deficits in the CNS.

CDGSH iron-sulfur domain 2 (CISD2), which is encoded by *CISD2* located at position 24 on the long arm of human chromosome 4 (4q24) [6], is one of the three members (CISD1–3) [7,8,9] of the human NEET family (with the proposed amino acid sequence asparagine-glutamate-glutamate-threonine) [10,11]. Additionally, CISD2 is characterized by the presence of a CDGSH iron-sulfur domain [12], which comprises CDGSH sequence motifs and [2Fe-2S] clusters connected by the coordinates of 3-cysteine-1-histidine [13]. The structure of the CISD2 protein comprises homodimers with each monomer containing a CDGSH domain (class I NEET protein) [14]. The specific CDGSH motif is highly conserved in archaea, bacteria, plants, and humans [15].

The CISD2 protein is localized to the outer membrane of mitochondria (OMM) [16], endoplasmic reticulum (ER) membrane [7], and mitochondrion-associated ER membranes (MAMs) [17]. Previous studies have reported that CISD2 regulates mitochondrial redox reactions and mitigates mitochondrial overload and ROS toxicity by functioning as a transport conduit for labile iron [18,19] and calcium [20] across the ER, MAMs, mitochondria, and cytosol. Additionally, CISD2 exhibits cytoprotective activities against calcium excitotoxicity [21,22], apoptosis [23,24], OMM breakdown, and mitochondrial dysfunctions [25].

Previously, we particularly found that the expression of CISD2 was reduced upon exposure to insults, including injury [26,27,28] or the aging process [1]. Based on the above, we aimed to examine the diagnostic value of CISD2 in patients with CNS insult. The findings of this study revealed that CISD2 expression was decreased in patients with CNS insult (e.g., trauma to the brain and spinal cord and stroke). The decrease in CISD2 levels was significantly correlated with the severity of CNS insults, implicating CICD2 as a potential biomarker of CNS insult.

Moreover, we demonstrated that CISD2 exerts anti-inflammatory effects [29] by regulating the upstream elements of the peroxisome proliferator-activated receptor-β (PPAR-β)/IκB/NF-κB signaling pathway [26]. To explore whether the anti-inflammatory effects of CISD2 are beneficial in the treatment of patients with CNS injury, in vivo models for inhibiting and potentiating CISD2 functions were established. As shown in Figure 1, the expression levels of CISD2 were decreased in the injured spinal cord (left panel, left row), brain, blood, and cerebrospinal fluid (CSF) (right panel, left row), with a concomitant increase in the levels of proinflammatory mediators at these sites in mice with SCI-induced inflammation in the brain (left panel) as well as in patients with CNS injury (right panel). Treatment with neutralizing anti-CISD2 antibodies enhanced the inflammatory response in the injured spinal cord and brain (left panel, right row). Increase in CISD2 function through targeted temperature management (TTM, i.e., therapeutic hypothermia) in patients with CNS insult attenuated CSF and blood levels of inflammatory mediators (right panel, right row). As such, we highlight that the upregulation of CISD2, which exerts anti-inflammatory effects by functioning as an NF-κB antagonist, is expected to directly inhibit the inflammatory cascade in the CNS and mitigate inflammation-induced neurological sequelae in patients with CNS injury or disease.

## 2. Materials and Methods

### 2.1. Ethics Statement

All patients in the study cohort provided their written informed consent. The experimental protocols were approved by the Institutional Review Board (IRB) of Kuang Tien General Hospital (IRB Approval No.: KTGH 10626, 14 August 2017), and the experiments were conducted according to the guidelines of the Declaration of Helsinki.

### 2.2. Patients

In this study, 13 consecutive patients with CNS injury aged 18–77 years were recruited between August 2017 and August 2018. Hemorrhagic stroke, TBI, brain tumor, and SCI were observed in five, five, one, and two cases, respectively. The vital parameters of these 13 patients were stable at the time of hospital admission. The patients did not exhibit damage to any other systems. Out of the 12 surgically treated patients, 11 underwent emergent decompression surgery within 3 h of hospital admission and 1 underwent conventional brain tumor surgery. The clinical characteristics of the patients are presented in Table 1. Blood samples from 11 patients with CNS insult were collected during decompression surgery, whereas the sample from one conservatively treated patient (case 7, Table 1) was collected immediately after admission. CSF samples from eight patients were obtained during external ventricular drainage or lumbar drainage. Three patients (cases 2, 12, and 13) underwent TTM on day 2 post-surgery. CSF samples from these patients were collected after the completion of TTM. The severity of brain insults and SCI was quantified using the Glasgow Coma Scale (GCS) and the American Spinal Injury Association Impairment Scale (AIS) grade. Hypothermia treatment:

All patients who underwent decompression surgery were subjected to TTM within 24 h of surgery according to the published therapeutic hypothermia protocol with modifications [30,31]. To perform TTM, the patients in the intensive care unit were wrapped with cold water-circulating wrapping garments (Arctic Sun Temperature Management System; Medivance, Louisville, CO, USA) equipped with a surface cooling device. The core body temperature was measured with bladder catheters in all patients. Cooled saline solution was applied to achieve rapid induction of hypothermia. The target temperature was 35 °C, which was maintained for 120 h. The temperature of the circulating water in the Arctic Sun system was adjusted to maintain the target temperature. The rewarming procedure was performed at a rate of 0.05–0.1 °C/h. During TTM, the patients were intubated and adequately sedated. Anti-shivering protocol [32] was adopted in shivering cases.

### 2.3. Animals

Adult BLTW: CD1 (ICR) male mice aged 8 weeks and weighing 31–33 g were obtained from the BioLASCO Experimental Animal Center (Taiwan Co., Ltd., BioLASCO, Yilan, Taiwan). Male Sprague–Dawley rats with a bodyweight of 280–330 g were purchased from Academia Sinica, Taipei, Taiwan. The animals were housed in cages (five mice or two rats per cage) for at least 5 days after arrival at the laboratory. Additionally, the animals had free access to food and were maintained under a regular circadian cycle (12-h light/dark cycle). Rats with SCI were used to collect CSF and to investigate the potential mechanisms involved in injury-induced decreases in CISD2. Rats were subjected to CSF collection because of the easy access to the cisterna magna and foramen magnum. Mice with SCI were used for other animal studies. This study was performed according to the guidelines outlined by the Experimental Animal Laboratory and approved by the Animal Care and Use Committee at National Ilan University, Yilan, Taiwan (IACUC approval number: 103-31).

### 2.4. Spinal Cord Hemisection

An animal model of spinal cord hemisection was established as described previously [33]. Briefly, the animals were anesthetized with isoflurane and fastened to a stereotaxic apparatus (David Kopf Instruments, Los Angeles, CA, USA). Under a dissecting microscope, resection of the lamina was performed at the 9th to 10th thoracic vertebrae that did not disrupt the integral dura. To perform a spinal cord resection, the wire knife guide is located along a vertical plane, near the lateral surface of the spinal cord at the inferior thoracic level. The knife is medially rotated and lengthened for 1.5 mm. An elevation of 4.0 mm is applied to the guide to allow for spinal cord hemisection. The sham group underwent only laminectomy without a hemisection. The wound was closed with sutures in layers. Animals were kept on a heating pad at 36.5 °C to recover and placed on fasting for 3 h after the procedure. For postoperative care, the animals were administered subcutaneous saline for fluid supplementation. Following surgical procedures, the animals were delivered back to their preoperative residence environment.

### 2.5. Collection of CSF from Rats with SCI

CSF was collected from rats according to a previously described method with modifications [34]. Rats were anesthetized by intraperitoneal injection of Zoletil™ (Zoletil 50, Virbac, Carros, France) at 4 h post-laminectomy (sham control group) or spinal cord hemisection (SCI4 group) (Figure 2A, lower panel) (*n* = 3 in each group). The neck hair was removed, and the surgical field was disinfected with alcohol. The neck skin was removed to expose the translucent dura mater of the brain. CSF was collected using a 30 G injection needle inserted into the dura mater at a 30° angle. The collected CSF was transferred to a microcentrifuge tube and stored in a deep freezer (−80 °C).

### 2.6. Cytokine Antibody Array Assay

The cytokines in the CSF samples of the rats belonging to the sham operation and SCI groups were analyzed using a cytokine antibody array kit (catalog: ab133992, Abcam, Cambridge, MA, USA), following the manufacturer’s instructions. Briefly, the membrane was incubated with blocking buffer. Next, the array pools with 34 targets were incubated with 1 mL CSF diluted to 70 µg/mL overnight to allow the binding of the anti-cytokine antibodies that are fixed at the membrane surface. The membrane was washed thrice with washing buffer I and twice with washing buffer II (5 min/washing step). The array pools were incubated with 1 mL of 1× biotin-conjugated anti-cytokine antibodies for 2 h, followed by incubation with 2 mL of 1× horseradish peroxidase (HRP)-conjugated streptavidin for 2 h. The membrane was imaged using an ImageQuant^TM^ LAS 4000 (GE Healthcare Life Science, Marlborough, MA, USA), and the densities were quantified using ImageQuant TL software.

### 2.7. Experimental Groups of the Mouse SCI Model

As illustrated in Figure 2A (upper panel), the mice were divided into the following groups: sham operation control group, animals underwent laminectomy (Sham control group) (*n* = 15); anti-CISD2 antibody-treated sham operation group, sham-operated mice intravenously administered with anti-CISD2 antibody(ShamT group) (*n* = 12); vehicle-treated 1 h post-SCI group (SCI1 group), animals underwent spinal cord hemisection and were administered with physiological saline and the brain and spinal cord were harvested at 1 h post-SCI (*n* = 12); vehicle-treated 4 h post-SCI group (SCI4 group), animals underwent spinal cord hemisection and were administered with physiological saline and the brain and spinal cord were harvested at 4 h post-SCI (*n* = 15); anti-CISD2 antibody-treated 4 h post-SCI group (Sham T group), animals underwent spinal cord hemisection and were administered with anti-CISD2 antibody and the brain and spinal cord were harvested at 4 h post-SCI (*n* = 15); vehicle-treated 12 h post-SCI group (SCI12 group) (*n* = 15); anti-CISD2 antibody-treated 12 h post-SCI group (SCI12T group) (*n* = 15); vehicle-treated 24 h post-SCI group (SCI24 group) (*n* = 9); vehicle-treated 36 h post-SCI group (SCI36 group) (*n* = 12); anti-CISD2 antibody-treated 36 h post-SCI group (SCI36T group) (*n* = 12); and vehicle-treated 48 h post-SCI group (SCI48 group) (*n* = 6).

The brain and spinal cord of animals belonging to different treatment groups were harvested for messenger RNA (mRNA) and protein analyses. Reverse transcription quantitative polymerase chain reaction (RT-qPCR), Western blotting, and enzyme-linked immunosorbent assay (ELISA) were performed using six, three, and six mice from each group, respectively.

### 2.8. Administration of Anti-CISD2 Neutralizing Antibody in Mice

Mice belonging to the sham operation and anti-CISD2 antibody-treated groups were intravenously injected with anti-CISD2 antibody (100 μg/kg bodyweight) at different time points post-SCI. Furthermore, the mice in the SCI group were intravenously administered with physiological saline post-SCI.

### 2.9. Histology, Immunohistochemistry, and Cell Count

The second day following SCIs, rats were deeply anesthetized by isoflurane and thoroughly perfused with phosphate-buffered saline (PBS), followed by cold 4% paraformaldehyde in 0.15 M sodium phosphate buffer. The spinal cords were removed, cleared in xylene, and embedded in paraffin. Five-micrometer sections were cut with a sliding microtome at the center of spinal cord hemisection. For immunohistochemistry staining, sections for injured tissues of spinal cords were washed in PBS and incubated in 3% normal goat serum with 0.3% Triton X-100 in PBS for 1 hr. The free-floating sections were incubated at 4 °C with the antibodies as follows. Anti-Aconitase 1 (ACO1) (Catalog: HPA019371, Sigma-Aldrich, St. Louis, MO, USA); anti-immunoglobin binding protein (BiP) (Catalog: IR130-525, iReal Biotechnology, Hsinchu City, Taiwan); anti-C/EBP homologous protein (CHOP)(Catalog: IR129-522, iReal Biotechnology, Hsinchu City, Taiwan); and anti-CISD2 (Catalog: PA5-34545, Thermo Scientific, Waltham, MA, USA). Afterward, the sections were washed and then incubated in anti-rabbit HRP secondary antibody (Catalog: GTX224125-01, Genetex, Hsinchu City, Taiwan). To visualize the bound antibody, the sections were reacted with diaminobenzidine (DAB) as the chromagen. Cell counting was performed on every sixth section at the center of spinal cord hemisection stained with the above antibodies at a magnification of 200×. Only cells with clearly visible stain were counted. All data are presented as mean + SEM of four consecutive cell quantifications.

### 2.10. Cell Lines

The microglial cell lines as EOC 13.31 and BCRC 60,490 cells were from the Bioresource Collection and Research Center (BCRC, Hsinchu, Taiwan). Colony-stimulating factor-1 (CSF-1) concentration determines the proliferation of EOC 13.31 cells from the brains of 10-day-old female mice (*Mus musculus*). We cultured EOC microglia in conditioned medium consisting of 70% Dulbecco’s modified Eagle medium, 10% fetal bovine serum, and 20% LDMAC conditioned medium (BCRC 60489, mononuclear cell line of bone marrow from adult mice). Conditioned LDMAC medium, containing CSF-1 from LADMAC cells, was collected from the cultures and filtered through syringes of 0.2 mm before being added to the EOC cell cultures.

### 2.11. Experimental Grouping of In Vitro Microglial Cells

To elucidate the anti-inflammatory mechanism of CISD2, EOC microglial cells were divided into the following groups: scrambled short-interfering RNA (siRNA)-transfected control group (*n* = 9); control group, EOC cells (1 × 10^6^ in a 35-mm dish) stimulated with 500 ng/mL LPS (Escherichia coli 055:B5, L-2880, Sigma Chemical Co., St. Louis, MO, USA) (*n* = 9); and siCISD2-transfected group (*n* = 9), LPS-stimulated EOC cells transfected with *CISD2*-specific siRNA (siCISD2). Three different cell cultures in each group were used for RT-qPCR, Western blotting, and ELISA.

### 2.12. RNA Interference

*Cisd2* in the cultured microglial cells was knocked down using siCISD2. siRNA transfection was performed according to a previously described method with minor modifications [1,29,35]. EOC microglial cells were transfected with siCISD2 or scrambled RNA (Silencer ^®^ Pre-designed siRNA, Ambion, Austin, TX, USA) using Lipofectamine™ 2000 reagent (Invitrogen, Carlsbad, CA, USA). EOC cells were cultured for 5 days and randomly assigned to the experimental groups. Lipofectamine™ 2000 reagent and siRNA were individually incubated with plasma-free medium for 10 min. The transfected cells (2 × 10^6^ cells) were transferred to individual plates. At 5 h post-transfection, the medium containing Lipofectamine 2000 was replaced with a microglial culture medium and the cells were incubated for 7 h. The knockdown efficiency of siCISD2 was determined using RT-qPCR.

### 2.13. RT-qPCR Analysis

Total RNA was extracted from the cultured cells or animal tissues using TRIzol reagent (Invitrogen). The extracted RNA was reverse transcribed to complementary DNA using oligo-dT and SuperScript II reverse transcriptase (Invitrogen). RT-qPCR analysis was performed using the ABI StepOne sequence detector system (Applied Biosystems, Foster City, CA, USA) with SYBR Green. The expression level of the target gene was normalized to that of a housekeeping gene [36]. The primers used for RT-qPCR analysis are listed in Table 2.

### 2.14. Immunoblotting

Total protein was extracted from cultured EOC microglial cells or animal tissues in a lysis buffer containing 20 mM Tris-HCl, 0.1% sodium dodecyl sulfate, 0.8% NaCl, and 1% Triton-X 100. The proteins were resolved using a 12% gradient gel. The resolved proteins were electro-transferred to a nitrocellulose membrane. The membrane was blocked with a blocking reagent and incubated with the following primary antibodies at 4 °C for 12 h: anti-CISD2 (1:500; catalog: PA5-34545; Thermo Scientific, Waltham, MA, USA); anti-iNOS (1:2000; catalog: PA3-030A; Thermo Scientific); anti-Arg1 (1:5000; catalog: PA5-29645; Thermo Scientific); anti-GAPDH (1:500; catalog: #MAB374; Millipore, Billerica, MA, USA); and anti-α tubulin 4a (1:1000; catalog: GTX112141; Hsinchu, Taiwan); anti-β-actin (1:4000; Catalog: ab8227; Abcam). Next, the membrane was washed and incubated with goat anti-rabbit IgG HRP-conjugated secondary antibodies (catalog: #12-348, Millipore) for 1 h. Immunoreactive signals were detected using a chemiluminescence detection kit (Catalog: WBKLS0500, Millipore). The target bands were visualized and quantified using ImageQuant^TM^ LAS 4000 (GE Healthcare Life Science, Marlborough, MA, USA).

### 2.15. Detection of NF-κB p65 Activity

NF-κB p65 activity was examined using an NFκB p65 transcription factor assay kit (catalog: ab133112, Abcam), following the manufacturer’s instructions. Briefly, the nuclear proteins were extracted to examine the intracellular NF-κB p65 activity. The absorbance was detected at 450 nm using a spectrophotometer reader (Synergy HT BioTek, Winooski, VT, USA).

### 2.16. ELISA

The plasma and CSF levels of CISD2, IL6, and CRP in patients with CNS insults were examined using human CISD2 ELISA kit (catalog: MBS7244473, MyBiosource, San Diego, CA, USA), human IL-6 Quantikine ELISA kit (catalog: D6050, R&D Systems, Minneapolis, MN, USA), and human CRP Quantikine ELISA kit (catalog: DCRP00, R&D Systems), respectively, whereas the brain levels of Tnfa in SCI mice were examined using a DuoSet^®^ mouse TNF-α ELISA kit (Catalog: DY410, R&D Systems). Briefly, 300 μL human plasma or CSF samples or mouse brain tissue lysates diluted (1:100) in lysis buffer were subjected to the immunoassay, following the manufacturer’s instructions. The standards of the target protein were analyzed in duplicates, whereas the samples were analyzed in triplicates. At least three independent human samples and six independent mouse samples were used for the analysis.

### 2.17. Statistical Analyses

The differences in the mean levels of inflammatory mediators in the CSF of rats between two groups were compared using two-sample independent t-test. The differences in mRNA and protein expression levels among three or more independent groups were compared using one-way analysis of variance, followed by Newman–Keuls post-hoc test. Values are displayed as mean ± standard error of the mean. All statistical analyses were performed using two-sided tests, and an significance level was determined by 0.05. Analyses were conducted in the GraphPad Prism software version 5.0 (GraphPad Software, Inc, San Diego, CA, USA).

## 3. Results

### 3.1. Plasma and CSF Levels of CISD2 Were Decreased in Patients with CNS Insult and Negatively Correlated with the Severity of the Injury

Previous studies have reported that the expression levels of CISD2 are decreased in the cell injury model and animal models of SCI, and the downregulation of CISD2, which exerts anti-inflammatory effects, leads to inflammation and mitochondrial dysfunction [1,28,29]. This indicated that the expression level of CISD2 is negatively correlated with the degree of injury and that CISD2 is a potential biomarker for CNS injury. For clinically validating previous in vivo and in vitro findings, the plasma and CSF levels of CISD2 were analyzed in patients with CNS insult. Additionally, the correlation between CISD2 expression and inflammatory status was examined in these patients.

The plasma and CSF levels of IL6 and CISD2 in 13 patients with CNS insult, including 11 head injury cases (6 TBI cases, 4 CVA cases, and 1 brain tumor case), 2 SCI cases (1 case each of AIS grade A [complete] and grade B [sensory incomplete] injuries), and 3 healthy controls were examined. The samples were collected immediately after hospital admission or during emergency surgery; therefore, the specimens indicate the inflammatory status during acute CNS insult.

The plasma CISD2 levels were significantly lower in patients with CNS insult (*n* = 13) than in healthy controls (*n* = 3; * *p* < 0.05, Figure 1A), whereas the plasma IL6 levels were increased in patients with CNS insult compared with those in healthy controls (Figure 1B). Consistent with the findings of previous animal and cellular studies, the plasma levels of CISD2 in patients with SCI (*n* = 2) and head trauma (*n* = 11) were non-significantly decreased compared with those in healthy controls (Figure 1C). The plasma IL6 levels were increased in patients with SCI and head trauma compared with those in healthy controls (Figure 1D). Patients with head trauma (*n* = 11) were divided into severe and mild groups (GCS scores of 3–8 and 9–15, respectively) according to the severity of brain injury. The plasma CISD2 levels in the severe and mild groups were lower than those in the control group. In addition, the plasma CISD2 levels in the severe group were markedly lower than those in the mild group (Figure 1E). The plasma IL6 levels were higher in the severe group than in the mild group (Figure 1F). Thus, compared with patients with CNS injury, healthy controls exhibited increased plasma CISD2 levels and decreased plasma IL6 levels (Figure 1C–F).

The CSF samples were available for seven patients (four CVA and three TBI cases) who underwent brain decompression surgery. Patients with brain trauma were divided into severe (GCS score of 3; *n* = 3) and mild groups (GCS scores of 5, 6, 8, and 9; *n* = 4). Consistent with the results of plasma analysis, the CSF levels of CISD2 were lower (Figure 1G) and those of IL6 (Figure 1H) were higher in the severe group than in the mild group. The analysis of clinical data of patients with CNS insult revealed that the plasma and CSF levels of CISD2 were decreased, whereas those of IL6 were increased. These results were consistent with those of previous in vivo and in vitro studies [1,26,27,28]. The plasma and CSF levels of CISD2 were negatively correlated with injury severity, whereas the plasma and CSF levels of IL6 were positively correlated.

Linear regression analysis was performed to examine the correlation between plasma CISD2 and plasma IL6 levels in 13 patients with CNS insult (11 head trauma and 2 SCI cases). The plasma CISD2 levels were negatively correlated with the plasma IL6 levels (*r* = −0.7062, *p* < 0.01) (Figure 1I).

These findings indicate that CISD2 is a potential biomarker for CNS insult. This is because excessive proinflammatory responses induced by CNS injury may inhibit the activation of CISD2. Next, the protective mechanisms of CISD2 against inflammation and mitochondrial dysfunction in CNS injury were examined.

### 3.2. Mouse Model of SCI Exhibited Inflammation in the Injured Spine and the Brain

**Dynamic** **1.**
*SCI mouse model exhibited decreased Cisd2 expression and elevated proinflammatory response in the spinal cord.*


Clinical studies have indicated that SCIs lead to cognitive decline [37,38]; depressive mood [39]; depression [40]; and neurodegenerative diseases, such as Alzheimer’s disease (AD) [41] and Parkinson’s disease (PD) [42]. The SCI-induced production of proinflammatory cytokines and neurotoxic molecules promotes brain inflammation [43]. This study demonstrated that SCI-induced inflammatory mediators are distributed throughout the brain and spinal cord through the circulation of CSF.

To induce SCI, the spinal cord of mice was subjected to hemisection at T9–T10. The mRNA expression levels of *Cisd2* and *Tnfa* in the central regions of the spinal cords of SCI mice were examined using RT-qPCR. Consistent with the results of previous in vivo and in vitro studies, the spinal cord levels of *Cisd2* mRNA in the 1 h (** *p* < 0.01), 24 h (** *p* < 0.01) and 48 h (** *p* < 0.01) post-SCI groups were significantly decreased compared with those in the sham operation group (Figure 2B). The spinal cord levels of *Tnfa* mRNA were significantly higher in the 1 h (*** *p* < 0.001) and 24 h (** *p* < 0.01) post-SCI groups than in the sham operation group (Figure 2C).

The results of RT-qPCR were consistent with those of Western blotting. As shown in Figure 2D–E, the spinal cord levels of the Cisd2 protein were significantly decreased (* *p* < 0.05) and the spinal cord levels of Nos2 protein were significantly increased (* *p* < 0.05) in the 24 h post-SCI group compared with those in the sham operation group.

**Dynamic** **2.**
*Proinflammatory mediators were detected in the CSF of SCI rats.*


Next, this study examined the role of proinflammatory response in the injured spinal cord in eliciting inflammation in the brain post-SCI. We hypothesized that CSF, which can circulate in the subarachnoid space around the brain and spinal cord, transports proinflammatory mediators released from the injured spinal cord to the brain.

To verify this hypothesis, the CSF of rats belonging to the sham operation and SCI groups were subjected to the cytokine protein array assays. The expression levels of proinflammatory mediators in the CSF collected at 4 h post-SCI from the sham operation and SCI groups were determined.

Figure 2F shows the levels of cytokines (indicated based on the intensity of the dots) in the CSF of the sham control (top left panel) and SCI groups (top right panel). The relative positions of these mediators in the cytokine protein arrays are shown in the panels below (Figure 2F).

As shown in Figure 2G, the CSF levels of Ngf (1.55 ± 0.02-fold higher, ** *p* < 0.01), Cxcl1 (3.17 ± 0.46-fold higher; *** *p* < 0.001), Faslg (1.23 ± 0.06-fold higher, ** *p* < 0.01), Ifng (1.32 ± 0.06-fold higher, ** *p* < 0.01), Il6 (1.28 ± 0.06-fold higher, * *p* < 0.05), Il13 (1.40 ± 0.12-fold higher, * *p* < 0.05), Lep (1.40 ± 0.04-fold higher, ** *p* < 0.01), Cxcl5 (1.53 ± 0.22-fold higher, * *p* < 0.05), Sell (1.67 ± 0.06-fold higher, ** *p* < 0.01), Ccl2 (3.22 ± 0.19-fold higher, ** *p* < 0.01), Ccl3 (2.15 ± 0.75-fold higher, * *p* < 0.05), Mmp8 (3.45 ± 0.21-fold higher, *** *p* < 0.001), Pdgfa (1.39 ± 0.11-fold higher, * *p* < 0.05), Tnfa (1.84 ± 0.34-fold higher, ** *p* < 0.01), and Vegfa (1.55 ± 0.16-fold higher, ** *p* < 0.01) in the 4 h post-SCI group were significantly higher than those in the sham operation group. These findings indicate that SCI-induced proinflammatory mediators were transferred to the brain through the CSF.

**Dynamic** **3.**
*Downregulated Cisd2 along with promoted inflammatory response in mouse brain as early as 1 h following spinal cord hemisection injury.*


A previous study reported the activation of astrocytes and the release of proinflammatory cytokines in the olfactory bulb (OB) at 8 h post-SCI [44]. An impaired sense of smell may be an early sign of various neurological disorders, such as PD [45], AD [46], stroke [47], and major depression disorder [48,49]. Interestingly, inflammation is observed in the brain upon CNS damage even in cases wherein the lesion is at distant sites, such as the spinal cord.

The brain tissue was excised from the sham operation and 1 and 4 h post-SCI groups. The mRNA expression levels of *Cisd2* and *Tnfa* in the brain tissues of the sham operation and 1 and 4 h post-SCI groups were examined using RT-qPCR. The brain levels of *Cisd2* mRNA were significantly decreased in the 1 (* *p* < 0.05) and 4 h post-SCI groups (** *p* < 0.01) compared with those in the sham operation group (Figure 3A). The brain levels of *Tnfa* mRNA were significantly increased in the 1 (*** *p* < 0.001) and 4 h (*** *p* < 0.001) post-SCI groups (Figure 3B). Additionally, the brain levels of *Tnfa* mRNA in the 4 h post-SCI group were significantly higher than those in the 1 h post-SCI group (*** *p* < 0.001; Figure 3B). This indicated that SCI rapidly induces brain inflammation. The results of ELISA were consistent with those of RT-qPCR analysis. The brain levels of Tnfa in the 1 (*** *p* < 0.001) and 4 h (*** *p* < 0.001) post-SCI groups were significantly higher than those in the sham operation group (Figure 3C). Additionally, the brain levels of Tnfa in the 4 h post-SCI group were significantly higher than those in the 1 h post-SCI group (** *p* < 0.001; Figure 3C).

Thus, CNS lesion due to spinal injury leads to the downregulation of CISD2 and the activation of local inflammatory responses in the spinal cord. SCI-induced proinflammatory mediators enter the brain through the CSF, which leads to the downregulation of CISD2 and the pathological activation of the brain inflammatory response (as early as 1–4 h post-injury). The expression of CISD2 is consistently decreased after injury. Thus, CISD2 can be a potential biomarker for CNS insult. Figure 3D shows the induction of inflammation in the brain after SCI with concomitant decrease in CISD2 expression in the injured spinal cord.

### 3.3. Inhibition of CISD2 Function Exacerbated Inflammatory Responses in the Brain and Spinal Cord of SCI Mice

Previous studies have reported the anti-inflammatory activities of CISD2 [1,26,28,29]. In this study, the downregulation of Cisd2 in the brain and spinal cord of SCI mice was associated with a concomitant upregulation of inflammatory mediators, which may exacerbate the CNS damage. To examine the causal correlation between CISD2 and injury-induced proinflammation in the CNS, mice were treated with anti-CISD2 antibodies to decrease the expression levels of Cisd2 in the brain and spinal cord of SCI mice. The sham operation and SCI groups were intravenously injected with anti-CISD2 antibody (100 μg/kg bodyweight) or physiological saline (vehicle group). The spinal cord levels of *Cisd2* and *Tnfa* in the 4, 12, and 36 h post-SCI groups were examined using RT-qPCR. The downregulation of Cisd2 was confirmed at the indicated time points in the spinal cord (**p* < 0.05; RT-qPCR; Figure 4A) and brain (** *p* < 0.01 and *** *p* < 0.001, respectively; Western blotting; Figure 4C) of the anti-Cisd2 antibody-treated SCI groups.

Cisd2 downregulation promoted inflammatory responses in the injured spinal cord and brain of SCI mice. The spinal cord levels of *Tnfa* mRNA in the anti-CISD2 antibody-treated sham operation (* *p* < 0.05) and 4 h (** *p* < 0.01), 12 h (* *p* < 0.05), and 36 h (*** *p* < 0.001) post-SCI groups were higher than those in the vehicle-treated sham operation and 4, 12, and 36 h post-SCI groups, respectively (Figure 4B).

Additionally, the brain levels of *Tnfa* mRNA in the sham operation and 4, 12, and 36 h post-SCI groups treated with a vehicle or anti-CISD2 antibody were examined using RT-qPCR. The brain levels of *Tnfa* mRNA were increased in the anti-CISD2 antibody-treated sham operation and 4 h (** *p* < 0.01), 12 h (* *p* < 0.05), and 36 h (** *p* < 0.01) post-SCI groups compared with those in the vehicle-treated sham operation and 4, 12, and 36 h post-SCI groups, respectively (Figure 4D). The effect of *Cisd2* downregulation on Tnfa protein expression in the brain of SCI mice treated with anti-CISD2 antibody was examined using ELISA. The brain levels of Tnfa in the anti-CISD2 antibody-treated sham operation (** *p* < 0.01) and 4 h (*** *p* < 0.001), 12 h (*** *p* < 0.001), and 36 h (** *p* < 0.01) post-SCI groups were significantly higher than those in the vehicle-treated sham operation and 4, 12, and 36 h post-SCI groups, respectively (Figure 4E).

The results of Cisd2 inhibition experiments demonstrated that Cisd2 exerts anti-inflammatory effects in vivo and that the CNS insult-induced downregulation of Cisd2 in the brain or spinal cord may lead to pathological neuroinflammation.

### 3.4. Cisd2 Suppression Exacerbated LPS-Induced Inflammation in EOC Microglial Cells

The results of in vivo study indicated that injury-induced CISD2 downregulation promotes inflammatory responses, which was further examined using EOC microglial cells. The aberrant activation of microglial cells is reported to activate inflammatory cascades that mediate the pathogenesis of various neurological diseases [50,51,52]. EOC microglial cells were stimulated with LPS (500 ng/mL) to induce inflammation and mimic the in vivo inflammatory response in the CNS.

As shown in Figure 5A, the *Cisd2* mRNA levels in the LPS-treated scrambled RNA-transfected cells were significantly decreased compared with those in the scrambled RNA-transfected cells (** *p* < 0.01). Additionally, the *Cisd2* mRNA levels in the siCISD2-transfected LPS-treated cells were significantly decreased compared with those in the scrambled RNA-transfected LPS-treated cells at 8 h post-treatment (*** *p* < 0.001), indicating the knockdown efficiency of siCISD2.

The mRNA expression levels of microglial cell-induced proinflammatory mediators, such as *Tnfa* (*** *p* < 0.001; Figure 5B) and *Ilb* (* *p* < 0.05; Figure 5C), in the LPS-stimulated scrambled RNA-transfected cells, were significantly higher than those in the scrambled RNA-transfected cells at 8 h post-LPS treatment. The expression levels of the LPS-induced proinflammatory cytokines *Tnfa* (** *p* < 0.01; Figure 5B) and *Ilb* (*** *p* < 0.001; Figure 5C), which are M1 microglial cell markers, in the siCISD2-transfected LPS-treated microglial cells were significantly increased compared with those in the scrambled RNA-transfected LPS-treated cells. The mRNA (*** *p* < 0.001; Figure 5D) and protein (* *p* < 0.05; Figure 5E) expression levels of Arg1, an M2 microglial cell marker, in the siCISD2-transfected LPS-treated cells, were lower than those in the scrambled RNA-transfected LPS-treated cells (representative immunoblot shown in the upper panel; Figure 5E) at 8 h post-LPS treatment. This indicated that LPS-induced injury downregulates the expression of Cisd2, which promotes the polarization of microglial cells toward the M1 phenotype and inhibits the polarization toward the M2 phenotype.

ELISA was performed to verify the effect of *Cisd2* knockdown on the DNA-binding activity of NF-κB p65 subunit in EOC microglial cells. The DNA-binding activity of NF-κB p65 in the scrambled RNA-transfected LPS-treated cells was significantly higher than that in the scrambled RNA-transfected cells at 8 h post-LPS treatment (* *p* < 0.05; Figure 5F). Additionally, the DNA-binding activity of NF-κB p65 in the siCISD2-transfected LPS-treated cells was higher than that in the scrambled RNA-transfected LPS-treated cells (** *p* < 0.01; Figure 5F). The results of CISD2 function suppression experiments demonstrated that CISD2 exerted in vitro anti-inflammatory effects in the LPS-treated microglial cells. The anti-inflammatory mechanism of CISD2 is summarized in Figure 5G. The downregulation of CISD2 suppresses the CISD2-mediated inhibition of IκB degradation (by activating PPAR-β), which promotes the nuclear translocation of NF-κB P65. In the nucleus, NF-κB P65 promotes the expression of proinflammatory genes, such as *TNFA* and *IL1B* (Figure 5G).

These findings indicate that the expression level of CISD2 was negatively correlated with injury severity and that the anti-inflammatory CISD2 protects against injury-induced inflammation, which adversely affects neural homeostasis.

Next, we would like to ask whether the defect in CISD2 is possibly specific to this protein itself or caused by a general defect in Fe-S metabolism, and whether the unfolded protein response (UPR) is involved in this situation as CISD2 is located in the ER. A spinal cord resection model in rats was conducted to explore potential factors for injury-driven decline in CISD2. As shown in Figure 5, IHC staining of CISD2 demonstrated significant loss in the SCI group after hemisection (Figure 5H), versus sham control (Figure 5L). After spinal cord hemisection, in the presence of reduced CISD2, the spinal cord lesions showed enhanced IHC staining for ACO1 (marker of Fe-S cluster biogenesis [53], Figure 5I), BiP (Figure 5J), and CHOP (Figure 5K) (marker of UPR [54]), as compared to sham controls (Figure 5M–O, respectively). For Figure 5H-O, the protein pattern in the dotted box (2400×) is a 12-fold magnification of the protein in the solid box (200×). CISD2, ACO1, CHOP, and BIP expression were analyzed by one-way ANOVA. The results indicated that significant differences for the sham control and SCI group occurred in CISD2 [F (1, 6) = 7.02, *p* < 0.05, Figure 5P], ACOI [F (1, 6) = 105.07, *p* < 0.05, Figure 5Q], and CHOP expressions [F(1, 6) = 5.36, *p* = 0.06, Figure 5S]. Though there was a trend towards elevated protein, there were no significant differences for BiP between the sham and SCI groups [F(1, 6) = 3.49, *p* > 0.05, Figure 5R]. These findings imply that CISD2 deficits after injury are likely to indicate the possibility that iron homeostasis could be altered, and are possibly involved in ER stress and associated UPR.

### 3.5. Potentiation of CISD2 Function Using TTM Increased CISD2 Expression and Decreased the Plasma and CSF Levels of Inflammatory Mediators in Patients with CNS Injury

TTM is reported to exert anti-inflammatory effects and enhance the microglial polarization toward the M2 phenotype in animal models [55,56]. Previously, we induced hypothermia in rats with spinal cord hemisection using continuous cryogenic spray cooling (CSC). CSC-mediated hypothermia mitigated injury-induced CISD2 downregulation and inhibited astrocyte activation, glial cell-mediated neuroinflammation, cellular apoptosis, and neuronal death [27]. Previous studies demonstrated that CISD2 exerts anti-inflammatory effects by functioning as an NF-κB antagonist and inhibiting injury-induced inflammatory cascades [26,29]. Thus, hypothermia-induced CISD2 upregulation may mitigate neuroinflammation and apoptosis.

The plasma and CSF levels of CISD2 and inflammatory mediators, such as IL6 and CRP, were examined in three patients with CNS injury (cases 2, 12, and 13 in Table 1; 1 case each of TBI, CVA, and SCI [AIS-A (complete)]). These three patients were subjected to TTM after decompression surgery. Hypothermia set in <30 h post-decompressive operation and was maintained for 5 days with a target temperature of 35 °C, followed by slow rewarming (0.25 °C/h).

In these three patients with CNS injury, the plasma (Figure 6A) and CSF (Figure 6C) levels of CISD2 after TTM were significantly higher than those before TTM. Additionally, TTM significantly mitigated the injury-induced upregulation of plasma CRP levels (Figure 6B) and CSF IL6 levels (Figure 6D).

Thus, therapeutic hypothermia exerted anti-inflammatory effects by increasing the levels of CISD2. These findings indicate that CISD2 is a biomarker for the anti-inflammatory effect of therapeutic hypothermia in patients. The results of CISD2 potentiation studies were consistent with those of in vitro and animal studies and demonstrated that CISD2 exerts anti-inflammatory effects in the CNS.

These findings reveal that the extent of neural damage can be monitored based on the expression levels of CISD2. To mitigate injury-induced inflammation, the expression level of CISD2 must be increased, indicating that CISD2 is a potential prognostic marker for CNS injury.

## 4. Discussion

Damage to the brain or spinal cord must not be managed as an individual disease in a clinical setting but as a CNS disease. Inflammatory mediators, such as TNFA, are reported to enter the blood–spinal cord barrier after SCI [42,57]. TNFA and various SCI-driven proinflammatory mediators [58] accumulate in the CSF, which circulates between the spinal cord and brain. TNFA disrupts the integrity of the blood–brain barrier (BBB), which results in enhanced permeability of the BBB and inflammation in the hippocampus of mice with depression [59]. In rodent models of SCI, neuroinflammation is mediated by the activation of microglial cells in the thalamus, hippocampus, and cerebral cortex [60]; the production of proinflammatory cytokines/neurotoxic molecules [43]; and the impaired secretion of brain-derived neurotrophic factor (BDNF) [61], which affects neuronal plasticity and promotes neurodegeneration in the brain. Clinical observations have indicated that SCI promotes cognitive impairment [37,38,39], which leads to the development of neurodegenerative diseases, such as AD [41] and PD [42]. Conversely, lesions in the brain can promote dysfunctions in the downstream systems. One study reported that corticospinal tract atrophy and motor neuron loss in the anterior horn of the spinal cord in a rat fluid percussion injury model of TBI were higher than those in the sham operation group at 12 weeks post-TBI [62].

Previously, we transfected the SH-SY5Y neuron-like and EOC microglial cells with siCISD2 to knock down the expression of *CISD2* and demonstrated the anti-inflammatory properties of CISD2. The knockdown of *CISD2* enhanced the secretion of inflammatory mediators (NOS2 and the chemokine regulated on activation, normal T cell expressed and secreted) and led to mitochondrial dysfunction, including decreased mitochondrial membrane potential (Δψm), enhanced ROS release, and ultimate cellular apoptosis in SH-SY5Y cells [1,28]. Transfection of EOC microglial cells lines with siCISD2 augmented the proinflammatory M1 phenotype (TNFA, IL1B, NOS2, and COX2) and downregulated the anti-inflammatory M2 microglia phenotype (ARG1, YM1, and IL10) [29]. Additionally, we found that CISD2 acts on the upstream components of the PPAR-β. Data showed that siCISD2-transfected cells exhibited decreased expression of PPAR-β [26] and augmented the DNA-binding activity of NF-κB P65 subunit [29]. More detailed, PPAR-β has been shown to stabilize IκB and prevent it from degradation, which in turn suppresses NF-κB signaling pathway [63]. As a consequence, CISD2 acts through the PPAR-β/IκB/NF-κB signaling pathway to exert its anti-inflammatory effects.

OB, which is the sentinel of neurodegeneration in the brain [45,64,65], mediates the olfactory projection among various functional nuclei [48,49,66] and stores neural stem cells (NSCs) [67]. Lesions at distant sites, such as the spinal cord [44] or other peripheral sites, increase the vulnerability of OB to injury [68]. OB injury decreases the signal transduction of olfactory projection and impairs neurogenesis, which leads to dysfunctional neurotransmission [65,69] and impairment of cognition and memory [70]. Pathological alterations in the OB post-SCI or injury at the remote peripheral sites are considered early signs of neurodegeneration in the brain. The infiltration of peripheral immune cells, such as lymphocytes, neutrophils, macrophages, and plasma cells, and the accumulation of proinflammatory cytokines in the OB were observed in rats intravenously administered with LPS at 6 h post-LPS [68]. Pathological changes in the OB were observed in mouse models of SCI at 8 h post-SCI. The SCI-induced micropathologies in the OB included reactive astrocyte-driven neuroinflammatory response, olfactory dysfunction, and dysregulated generation of NSCs and associated neurotrophic factors, such as BDNF [44]. In this study (Figure 3B,C), the mRNA levels of *Tnfa* in the brain (not restricted only to the OB) were increased at 1 h post-spinal cord hemisection injury in mice. This indicates that trauma may rapidly activate the proinflammatory cascades in the entire CNS and subsequently promote inflammation in the brain.

The above-mentioned phenomenon was confirmed in our results. The decreased CISD2 expression post-CNS injury results in the downregulation of CISD2-mediated anti-inflammatory responses in the CNS and the exacerbation of proinflammatory responses in the CNS. The brain levels of *Cisd2* mRNA were decreased (Figure 3A), whereas the brain levels of Tnfa mRNA (Figure 3B) and protein (Figure 3C) were significantly increased in the 1 and 4 h post-SCI groups compared with those in the sham operation group. In SCI-driven inflammatory brain, the expression level of Cisd2 gradually decreased from 1 h to 4 h post-injury. However, the expression of Tnfa significantly increased from 1 h to 4 h post-injury. Thus, spinal injury-induced brain inflammation decreases the expression of CISD2 and consequently contributes to the degeneration of the brain. Accordingly, investigation of injury biomarkers or exogenous/endogenous anti-inflammatory mechanisms, including CISD2-dependent mechanisms, is mandatory for the theranostics of CNS injury and disease.

Previous studies reported that the expression of CISD2 is decreased in cell or animal models of inflammation-induced injury [28] or aging [1]. The expression of CISD2 was decreased upon exposure to various insults, including injury or the aging process. For example, the expression of CISD2 is downregulated in mice [26,28] or rats [27] after spinal cord hemisection, lipopolysaccharide (LPS)-stimulated astrocytes [26,28], and SH-SY5Y neuron-like cells [28], and in 104-week-old mice [1] and astrocytes after long-term culture (35 *DIV*) [1]. Hence, in this study, we reasonably hypothesized that CISD2 is a potential biomarker for the degree of damage in the CNS and that CISD2 exerts anti-inflammatory effects in the CNS. The expression of CISD2 was examined in patients with CNS insults. The plasma and CSF levels of CISD2 were lower in patients with CNS insults than in the healthy controls. Additionally, the plasma and CSF levels of CISD2 were negatively correlated with the severity of CNS insults. A linear regression model was used to examine the correlation between the plasma levels of CISD2 and IL6 (a proinflammatory cytokine that is positively correlated with the severity of CNS injury). The correlation coefficient was −0.7062, which indicated a strong negative correlation [71]. Therefore, CISD2 was demonstrated to be a biomarker for CNS insult.

The results of experiments on the inhibition and potentiation of CISD2 functions demonstrated the anti-inflammatory activities of CISD2, which will enable the application of CISD2 in neuroscience research. In this study, CISD2 was downregulated using siCISD2 in LPS-stimulated EOC microglial cells (to mimic microglial cell-induced inflammation) or the anti-CISD2 neutralizing antibodies in mice (to mimic SCI-driven inflammation of the brain and spinal cord) (Figure 4 and Figure 5). Treatment with LPS shifted the polarization of microglial cells toward the inflammatory M1 phenotype. Additionally, *Cisd2* knockdown exacerbates the proinflammatory responses in both in vivo and in vitro models. The function of CISD2 in patients with CNS insults was potentiated using TTM. TTM downregulated the plasma and CSF levels of proinflammatory mediators (Figure 6). These findings provide more evidence of the anti-inflammatory properties of CISD2. Moreoveer, CISD2 can regulate the upstream proinflammatory molecules, such as IκB and NF-κB, in the cascade of proinflammatory responses [26,29]. Thus, strategies that increase CISD2 expression levels can directly achieve widespread and effective anti-inflammatory effects. Previous studies have reported various strategies, including *Cisd2* overexpression in AD mice [72], experimental cryogen spray cooling in SCI rats [27], and administration of curcumin [1,28] and *Momordica charantia* Linn. var. Abbreviata Ser. [26], to increase the expression of CISD2 and enhance its neuroprotective effects. In this study, the expression of CISD2 was increased in patients with CNS insults who underwent TTM.

This study has several limitations. This study included a relatively small number of patients, especially the controls, in spite of the fact that the data obtained is statistically significant. Further quantification studies with a wide range of patients should be conducted to consider CISD2 as a potential biomarker for clinical implementation. Second, our findings in cells, animals, and clinical patients suggest that CNS insult downregulates CISD2. However, the mechanism underlying injury-induced CISD2 downregulation has not been thoroughly elucidated. As a family of Fe-S proteins, CISD2 protects against ER or mitochondrial stress by mediating the transport and homeostasis of labile iron, calcium, and ROS across the ER, mitochondria, and MAMs [18,20,24]. The current study found that injury-driven CISD2 depletion was probably related to ER stress and part of the UPR. Further studies are warranted in the future to determine whether the protein folding and modification of CISD2 is influenced after injury, to further determine if it is involved in the UPR process, and to detail the depletion mechanism. Furthermore, ACO1, the Fe-S bifunctional protein, acts as catalytically aconitase enzyme in the the Krebs cycle [73] or mRNA binding protein in regulating iron uptake and utilization [74] at respective high or low iron levels. Because of the increased levels of ACO1 in SCI of our data, we suggest that CISD2 deficits after injury are possibly related to alterations in iron homeostasis; however, whether the deregulation of iron homeostasis is also involved in SCI and its potential relationship with reduced CISD2 is a concern to be further investigated. Nevertheless, the results of this study showed that the expression of CISD2 was negatively correlated with the severity of the disease. Accordingly, CISD2 is a biomarker of CNS injury. The novel findings of this study provide a basis for the development of CISD2-dependent diagnostic strategy for the nof CNS insults and will enable the establishment of CISD2-based anti-inflammatory drug intervention in clinical practice.

## 5. Conclusions

We previously reported that CISD2 is downregulated after injury. Consistently, this study demonstrated that CISD2 is a biomarker for CNS insult and that CISD2 expression is negatively correlated with the severity of the disease. The CNS is susceptible to external influences. Inflammatory responses were observed in the brain at 1 h post-SCI. Thus, CISD2-mediated neuroprotective effects are suppressed in the CNS after injury. The findings of this study will enable the development of novel therapeutic strategies for CNS injury, which may involve the upregulation of anti-inflammatory CISD2.

## Data Availability

The data used to support the finding of this study are included within the article.

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
