# Peer review of "Anti-Inflammatory CDGSH Iron-Sulfur Domain 2: A Biomarker of Central Nervous System Insult in Cellular, Animal Models and Patients"

_biomedicines, 2022, doi:10.3390/biomedicines10040777_

Round 1

Reviewer 1 Report

Manuscript ID: biomedicines-1648760

Title: Anti-inflammatory CDGSH iron-sulfur domain 2: A biomarker of central nervous system insult in cellular, animal models and patients

To determine the relationship between CDGSH downregulation and Fe-S biosynthesis / UPR pathways the authors choose a battery of protein markers involved in these processes and analyzed them by immunohistochemistry. The results regarding the UPR markers are clear as the upregulation of all markers tested point that probably the downregulation of CDGSH is also associated to ER stress. This is properly stated in the discussion. To determine whether Fe-S cluster biogenesis pathways are affected in the SCI the authors choose ACO1 as a marker. Aconitase 1 is an Fe-S bifunctional protein.  Depending on iron levels the protein can function either as an aconitase enzyme in the TCA cycle or as an mRNA binding protein. If iron levels are high the protein contains Fe-S centers and is catalytically active converting citrate to isocitrate in the Krebs cycle. When iron levels are low Aco1 regulates iron uptake and utilization by binding to iron-responsive elements located on the mRNA of genes involved in iron homeostasis. The authors detected extremely increased levels of Aco1 in SCI and stated that Fe-S synthesis is not affected. In my opinion these results do not necessarily indicate that Fe-S cluster biogenesis is unaffected (although is probably true) but would more likely indicate the possibility that iron homeostasis could be altered.

The authors conclude that “These findings imply that CISD2 deficits after injury are probably not due to general Fe-S-dependent activity impairment and are possibly involved in ER stress and associated UPR “. Taken into account bifunctional role of Aco1 I would suggest to be cautious and this issue should be considered and properly discussed in the manuscript, since the significant increase of Aco1 levels in SCI raises the question whether the deregulation of iron homeostasis is also involved in SCI.

Author Response

Point to point response as attached file

Reviewer 2 Report

The revised mansucript is acceptable in its present form.

Author Response

Point to point response as attached file

This manuscript is a resubmission of an earlier submission. The following is a list of the peer review reports and author responses from that submission.

Round 1

Reviewer 1 Report

I have recommendations on form.

Title : « in human, cellular and animal models ». Why are the words in this order ? It does not seem logical to me, unless « human » relates to a model ?

Too many results in the introduction section, please be more concise. Also, figure 1 is placed before the end of the introduction section.

There are image resolution issues to correct (Scheme 1,  fig 3D, fig 5G)

Page 14 . Title of section 3.2.3 : what does  « Dynamic 3 » stand for ?

Table 1 is hardly readable and should be rebuilded .

No line numbers…

Change PPARB and NFKB to PPAR b or beta and NF-k (or kappa) B. What is IKBKB ?

PPAR beta role should be more discussed / explained, so should be its link with NF kappaB.

Author Response

Point to point response as attached file. 

Reviewer 2 Report

Review: biomedicines-1368622 

The manuscript entitled "Anti-inflammatory CDGSH iron-sulfur domain 2: A biomarker of central nervous system insult in human, cellular, and animal models” by  Kung and co-workers highlights the role CISD2 as a biomarker of CNS injury. The authors provide fundamental data in both, patients’ samples and animal and cellular models. In my opinion the manuscript is interesting.

However, since CISD2 is a protein localized in the mitochondrial membrane, endoplasmic reticulum and membranes associated to these organelles I suggest that some data regarding the status of these organelles is necessary to considerer the manuscript for publication.

1) The authors showed that plasma and CSF levels of CISD2 are decreased in patients with CNS injury, suggesting that CISD2 levels could be a biomarker for CNS insults (as stated in the manuscript title). The authors showed quantitative data but they should also provide a wide control range in order to consider CISD2 as a potential biomarker to be used in a clinical context. 

2) As CISD2 is involved in mitochondrial function, a characterization of the mitochondrial phenotype in cells and animal tissues used in the study should be appropriate (mitochondrial respiration, mitochondrial mass -citrate synthase-, morphology, mitochondrial ROS, …).

3) The fact that CISD2 is an Fe-S protein it will be relevant to check the levels of other Fe-S proteins to see whether a general reduction of this pathway could explain the low levels of CISD2. It is known that Fe-S biosynthesis defects lead to reduced levels of Fe-S proteins due to the lack of the cofactor. This information will be important to ensure that the defect of CISD2 is either specific of this protein or caused by a general defect of Fe-S metabolism, especially considering that CISD2 is involved in labile iron homeostasis. The analysis of aconitase activity ( a well known Fe-S dependent activity) could be useful.

4) Since CISD2 localized in also localized in the ER it should be interesting to test whether UPR is increased in the models used.

Author Response

(The authors gave the same response as above.)

Round 2

Reviewer 2 Report

Dear editors, 

The authors have addressed the major points of the revision. As not enough time was available to perform new experiments the authors have discussed and considered the suggestions by including new parapgraphs in the manuscript text.